# Comparison of the Efficacy and Safety of Trabeculectomy with Mitomycin C According to Concentration: A Prospective Randomized Clinical Trial

**DOI:** 10.3390/jcm10010059

**Published:** 2020-12-26

**Authors:** Bo Ram Seol, Sang Yoon Lee, Yu Jeong Kim, Young Kook Kim, Jin Wook Jeoung, Ki Ho Park

**Affiliations:** 1Department of Ophthalmology, Seoul National University College of Medicine, Seoul 08826, Korea; gorong20@hanmail.net (B.R.S.); forgotten100@gmail.com (S.Y.L.); md092@naver.com (Y.K.K.); neuroprotect@gamil.com (J.W.J.); 2Department of Ophthalmology, Veterans Health Service (VHS) Medical Center, Seoul 05368, Korea; 3Department of Ophthalmology, SNU Blue Eye Clinic, Seoul 08745, Korea; 4Department of Ophthalmology, Seoul National University Hospital, Seoul 03080, Korea; yjkimhappy@hanmail.net

**Keywords:** efficacy, mitomycin C, trabeculectomy, safety

## Abstract

(1) Background: Mitomycin C (MMC) is commonly used during trabeculectomy. However, there is no consensus on which concentration should be used. We aimed to compare the efficacy and safety of 0.2 mg/mL and 0.4 mg/mL of MMC in eyes undergoing trabeculectomy. (2) Methods: Thirty-six eyes (36 glaucoma patients) were randomized to undergo a trabeculectomy with 0.2 mg/mL or 0.4 mg/mL of MMC. The success rate was evaluated according to three criteria: (A) intraocular pressure (IOP) ≤ 18 mmHg and IOP reduction ≥ 20%; (B) IOP ≤ 15 mmHg and IOP reduction ≥ 25%; (C) IOP ≤ 12 mmHg and IOP reduction ≥ 30%. Cox’s proportional hazard model was used to identify the predictive factors for failure. Immunohistochemical procedures for matrix metalloproteinase (MMP) were performed on Tenon’s tissue. Bleb morphology was evaluated. Safety was assessed based on the incidence of complications. (3) Results: Of the 36 eyes, 19 underwent trabeculectomy with 0.2 mg/mL of MMC and 17 with 0.4 mg/mL. The success rates were 75%, 67%, and 47% at 6 months for criteria A, B, and C, respectively. There were no significant differences between the two groups. High MMP-9 staining and low preoperative IOP were associated with failure (hazard ratio (HR), 5.556; *p* = 0.033, and HR, 0.936; *p* = 0.033). Complications included hypotony in two eyes (6%), hyphema in one eye (3%), and choroidal detachment in one eye (3%). (4) Conclusions: Trabeculectomy with 0.2 mg/mL and 0.4 mg/mL of MMC showed similar IOP-control effects to those recorded in previous studies, along with a low rate of complications. There was no significant difference in efficacy or safety between the 0.2 mg/mL and 0.4 mg/mL MMC groups.

## 1. Introduction

A common cause of trabeculectomy failure is the formation of a subconjunctival scar due to the wound-healing reaction [1,2]. Mitomycin C (MMC) is commonly used to prevent cicatricial adhesion and enhance trabeculectomy success rates [3,4]. However, there is no consensus on which concentration should be used [5,6,7,8]. The concentrations of MMC used in trabeculectomy vary from 0.1 mg/mL to 0.4 mg/mL, depending on the surgeon. When high-concentration MMC is used, the effect of wound-healing inhibition is improved, but the possibility of side effects such as postoperative hypotony, avascular bleb, bleb leak, and endophthalmitis is increased [5,9,10]. Although several previous studies have evaluated MMC use in trabeculectomy, the efficacy and safety of MMC according to its concentration remain unclear [5,6,7,8,11,12,13,14].

In the histological aspect, matrix metalloproteinase (MMP) is important for subconjunctival scarring due to its interaction with proliferating fibroblasts inhibited by MMC during wound healing [15,16]. The degree of expression of MMP, which might be related to its effect, affects the result of surgery. Therefore, when determining whether to use MMC in trabeculectomy, its type and degree of expression in tissues should be considered. However, histologic analyses in cases of MMC-assisted glaucoma surgery are very rare [17,18].

Therefore, we started a prospective randomized clinical study to identify the efficacy and safety of MMC in trabeculectomy according to concentration. Additionally, we evaluated the risk factors for surgical failure, including MMP expression as a possible associated factor.

## 2. Experimental Section

### 2.1. Study Design

This was a prospective, double-blind, randomized, active-controlled, parallel-group study. Subjects were recruited from the patient population of Seoul National University Department of Ophthalmology from April 2015 to May 2016. After explaining the method, including the benefits and risks of the procedure, informed consent was obtained from all of the patients. The study was conducted in accordance with the Declaration of Helsinki and Consolidated Standards of Reporting Trials (CONSORT) statement. It was approved by the Institutional Review Board (IRB) of Seoul National University Hospital (1410-081-618) on 3 February 2015. This trial was registered at cris.nih.go.kr on 1 July 2019 (KCT0004108).

### 2.2. Subjects

Patients diagnosed with primary open-angle glaucoma (POAG), primary angle-closure glaucoma (PACG), pseudo-exfoliation glaucoma, or secondary glaucoma were screened. The inclusion criteria comprised an age older than 20 years and inadequate intraocular pressure (IOP) control despite maximally tolerated medical therapy. In addition, eyes with IOP less than 21 mmHg that received IOP-lowering medication but showed the progression of optic nerve damage or deterioration of visual field (VF) were included. Glaucomatous eyes were defined as eyes showing glaucomatous optic disc appearances including neuro-retinal rim thinning, notching, and/or retinal nerve fiber layer (RNFL) defects and corresponding glaucomatous VF defects, as confirmed by at least two consecutive VF examinations. Glaucomatous VF defects were defined as a cluster of ≥3 points with *p* < 0.05 on the pattern deviation map in at least one hemifield, including ≥ 1 point with *p* < 0.01; a pattern standard deviation (PSD) of *p* < 0.05; or glaucoma hemifield test result outside the normal limits with reliable VF test results (fixation loss <20%, false-positive errors <15%, and false-negative errors <15%) [19].

Exclusion criteria were previous intraocular surgery except for cataract operation and known allergy to MMC. Patients with cornea features that could affect IOP measurement, including keratoconus, history of penetrating keratoplasty, or refractive surgery, and retinal disease that could affect VF assessment, including non-glaucomatous optic neuropathy, diabetic retinopathy, or vascular occlusion, were also excluded. In addition, patients with thrombocytopenia or coagulopathy, those receiving phenytoin as a yellow fever vaccine or prophylactic agent, and fertile women who were pregnant or planned to become pregnant during the follow-up period were excluded. In one patient, for whom both eyes satisfied the inclusion criteria, the eye that had surgery first was included.

### 2.3. Preoperative Assessment

All patients underwent a baseline ophthalmologic examination before surgery including measurements of best-corrected visual acuity (BCVA), IOP measurement (by Goldmann applanation tonometry), corneal pachymetry (Pocket II Pachymeter Echo Graph; Quantel Medical, Clermont Ferrand, France), axial length (AXL) measurement (IOL Master; Carl Zeiss Meditec Inc., Jena, Germany), slit-lamp examination, gonioscopy, dilated fundus examination, color disc photography, red-free RNFL photography (Vx-10; Kowa Optimed, Tokyo, Japan), anterior-segment photography, and Humphrey Visual Field Analysis (Carl Zeiss Meditec Inc.; Dublin, CA, USA) using the Swedish interactive threshold algorithm with the 24–2 standard program. Age, sex, surgeon, type of glaucoma, presence of systemic disease, previous laser or operation history, central corneal thickness, AXL, BCVA, preoperative IOP (the final IOP before surgery), number of glaucoma medications, and VF indices were noted for all patients at baseline.

### 2.4. Randomization

Enrolled eyes were randomized for treatment with either 0.2 mg/mL or 0.4 mg/mL of MMC during the trabeculectomy. Patients were assigned to one of the MMC 0.2 mg/mL and 0.4 mg/mL group at a 1:1 ratio. Randomization using the block randomization method was done. A mixture of block sizes 4 and 6 was used. After the researcher applied for a random assignment to the Seoul National University Hospital Medical Research Cooperation Center, she was given a unique ID and then logged on to http://mrcc.snuh.org/ and entered the random assignment computerized system. Afterward, the necessary information for the random assignment (the selection exclusion criteria conformity information) was input and the assignment was received. Web-based randomization was conducted at the Medical Research Collaborating Center of Seoul National University and Seoul National University Hospital.

### 2.5. Surgical Procedure

All subjects underwent the same trabeculectomy procedure with the patient and surgeon blinded to the concentration of MMC. The trabeculectomy in every case was carried out by three surgeons (K.H.P., J.W.J., and Y.K.K.). After topical anesthesia with proparacaine 0.5%, the eye was draped. A corneal traction suture was performed with 6–0 silk, and a fornix-based conjunctival flap was formed at the superior limbus. Then, dissection of a 3 × 3 mm^2^ rectangular scleral flap of half-thickness was done. MMC was applied by 4–5 blocks of 2 × 2 mm sized cellulose sponges placed on the episclera over 2 h at a concentration of 0.2 mg/mL or 0.4 mg/mL for 2 min. After this procedure, MMC was washed with 30 mL of balanced salt solution. A full-thickness ostium was excised at the trabecular meshwork with a punch, and an iridectomy was done. Then, the scleral flap was sutured with 10–0 nylon at each corner. In addition, two limbal sutures at the edge of the flap and one mattress suture in the middle of the limbal wound were done with 10–0 nylon. As the last step, a subconjunctival dexamethasone injection was performed.

### 2.6. Postoperative Assessment

Postoperatively, all eyes received topical antibiotics 4 times daily (levofloxacin 0.5%) and topical steroid 4 times daily (prednisolone acetate 1%). The subjects were followed up at 1 day, 1 week, and 1, 3, and 6 months after surgery. A window of 7 days was allowed for the 1- and 3-month visits; a window of 14 days was permitted for the 6-month visit. The timing of laser suture lysis, needling, 5-fluorouracil (FU) injection, and bleb massage after surgery might have varied. The postoperative data included BCVA, IOP, number of glaucoma medications, bleb grading (evaluated by anterior-segment photography performed after 1 week postoperatively), and complications. The bleb morphology was evaluated by two glaucoma specialists (B.R.S. and S.Y.L.) using the Moorfields bleb grading system (MBGS) [20]. If the two disagreed, a third glaucoma specialist (J.W.J.) decided. The system scored the following seven different bleb parameters: central and maximal bleb area; bleb height; central, peripheral, and non-bleb vascularity; and presence of subconjunctival hemorrhage [20]. Hypotony was defined as an IOP of 5 mmHg or less at least 1 month after surgery [21,22].

### 2.7. Experimental Procedures

MMP staining was performed using a similar process to that of the previous study as follows [18]. An approximately 2 × 2 mm Tenon’s tissue sample was obtained from an area 2–3 mm posterior to the limbus at the beginning of the operation. The 36 tissue samples (from 36 patients) were fixed in neutral buffered 10% formalin for 24 h, after which they were embedded in paraffin wax and sectioned at 4 µm thickness. The samples were then examined with three antibodies according to a Envision+ Detection system (HRP/DAB+, DAKO, Glostrup, Denmark). The monoclonal antibodies used were anti-human MMP-1, -2, -3 and -9 (1 in 100; Calbiochem, Cambridge, MA, USA). The dewaxed sections were subjected to antigen retrieval using 20 μg/mL proteinase kinases in 10-min courses, and endogenous peroxidase activity was blocked by exposing the sections to 3% H2O2 for 10 min. The sections were incubated with primary antibody overnight at 4 °C. After washing with PBS, sections were stained using EnVision+ and counterstained with Mayer′s hematoxylin (DAKO). The cells expressing the MMPs acquired a brown-chestnut coloration in the cytoplasm, enabling their identification and quantification. Immunohistochemical staining was evaluated by two independent, masked observers (B.R.S. and S.W.L.). If the two disagreed, a third glaucoma specialist (J.W.J.) decided. The samples were graded into two groups based on the extent of staining, weak or strong.

### 2.8. Outcome Evaluations

Bleb grade score at 6 months postoperatively was the primary outcome measure. Bleb grade score at 1 day, 1 week, 1, 3, and 6 months postoperatively; IOP and number of glaucoma medications at 1 day, 1 week, 1, 3, and 6 months postoperatively; success rates and the predictive factors for surgical failure; and complications and additional procedures during the follow-up period were the secondary outcome measures. In other words, the difference between the two groups was analyzed in two aspects, efficacy and safety. The efficacy of surgery was analyzed according to changes in IOP, number of glaucoma medications, success rate, and bleb morphology. Safety was assessed based on postoperative complications. Additional procedures including laser suture lysis, needling, injection of 5-FU, and bleb massage were also evaluated. Surgical success was defined by reference to the criteria of previous studies [4,16,17]. The success of the surgery was defined by the criteria related to IOP with or without glaucoma medications and the lack of additional IOP-lowering surgery. The IOP-related criteria were as follows: (A) IOP ≤18 mmHg and ≥20% reduction of IOP from the preoperative IOP; (B) IOP ≤15 mmHg and ≥25% reduction of IOP from the preoperative IOP; (C) IOP ≤12 mmHg and ≥30% reduction of IOP from the preoperative IOP [6]. Time of failure was defined as the time of the first of two events: failure of the IOP-related criterion or additional IOP-lowering surgery.

### 2.9. Statistics

Kaplan–Meier survival analysis was used to evaluate the success of surgery, and *p-*values were obtained using a log rank test. In the survival analysis, the endpoint was defined as the time point at which the first progression was detected. The time when progress was detected was regarded as the end of follow-up. Cox’s proportional hazard model was used to assess the risk factors for failure. Univariate analysis was performed for each factor. Multivariate analysis was performed using the factors with *p* < 0.2 in the univariate analysis. Backward elimination was used to develop the final multivariable model, and adjusted HRs with 95% confidence intervals were calculated. The comparisons of IOP, number of glaucoma medications, bleb grading, and postoperative complications and additional procedures between two groups were performed using the Mann–Whitney test for continuous variables and the Fisher’s exact test for categorical variables. The statistical analysis was carried out using SPSS 18 for Windows (SPSS Inc., Chicago, IL, USA). A *p*-value of <0.05 was considered significant.

### 2.10. Sample Size

We calculated the number of study subjects according to the score for bleb central vascularity. In a previous study, the mean and standard deviation (SD) of the MBGS scores at 12 months after surgery in the MMC 0.2 mg/mL group were 1.80 and 0.67, respectively [23]. We assumed that the mean of the MMC 0.4 mg/mL group was 1 SD lower than the mean value of MMC 0.2 mg/mL, and the SD was 0.67. With 80% power and a type I error of 5%, the estimated sample size was 16. A sample size of 18 would have to be recruited for each group considering a 10% loss to follow-up.

## 3. Results

### 3.1. Subjects

Thirty-six patients undergoing trabeculectomy between April 2015 to May 2016 were included in this study. After trabeculectomy, they were followed up for 6 months. The study protocol according to the Consolidated Standards of Reporting Trials (CONSORT) statement is shown in Figure 1. The patients’ demographics and baseline characteristics are summarized in Table 1. Nineteen primary open-angle glaucoma (POAG) eyes, four primary angle-closure glaucoma (PACG) eyes, four pseudo-exfoliation glaucoma eyes, and nine secondary glaucoma eyes were examined. For 18 patients, 0.2 mg/mL MMC was used, and for 16 patients, 0.4 mg/mL was used, after randomization.

### 3.2. Changes of Intraocular Pressure and Number of Glaucoma Medications

The changes in intraocular pressure (IOP) and number of glaucoma medications are shown in Table 2. At the 6-month visit, the mean (standard deviation (SD)) IOP had decreased from 24.72 (8.64) mmHg before surgery to 12.88 (4.63) mmHg. The mean (SD) reduction of IOP was 12.50 (8.42) mmHg, and the mean (SD) percentage reduction of IOP was 44.98 (20.92). The mean (SD) number of glaucoma medications also had decreased at the 6-month visit. In terms of IOP, reduction in IOP, percentage reduction in IOP, and number of glaucoma medications, there were no statistically significant inter-group differences at any of the follow-up visits.

### 3.3. Success Rates and the Predictive Factors for Surgical Failure

Figure 2 plots the results of the Kaplan–Meier survival analysis. For criterion A, 14 of 19 (73.7%) and 13 of 17 eyes (76.5%) showed success in the 0.2 mg/mL and 0.4 mg/mL MMC groups, respectively, 6 months after surgery. For criterion B, 10 of 19 (52.6%) and 12 of 17 eyes (70.6%) showed success in the 0.2 mg/mL and 0.4 mg/mL MMC groups, respectively, 6 months after surgery. For criterion C, 8 of 19 (42.1%) and 9 of 17 eyes (52.9%) showed success in the 0.2 mg/mL and 0.4 mg/mL MMC groups, respectively, 6 months after surgery. For criteria A, B, and C, there were no statistically significant survival curve differences between the two groups (log-rank test, *p* = 0.847, 0.323, and 0.537, respectively). Univariate and multivariate Cox’s proportional hazard models were used to determine the predictive factors for surgical failure (Table 3). In the univariate analysis, low preoperative IOP was associated with failure according to success criteria A, B, and C (hazard ratio (HR), 0.858; *p* = 0.010, HR, 0.910; *p* = 0.018, HR, 0.936; *p* = 0.043, respectively). For criteria B, laser suture lysis and high-MMP-9 staining were also associated with failure (HR, 3.895; *p* = 0.038, HR, 5.556; *p* = 0.033). In the multivariate analysis, for criteria A, there were no factors associated with failure. For criteria B and C, high-MMP-9 staining and low preoperative IOP were risk factors for failure, respectively (HR, 5.556; *p* = 0.033, HR, 0.936; *p* = 0.033).

### 3.4. Bleb Morphology

Table 4 shows the changes in bleb morphology during the follow-up period. In the bleb height evaluation, the MMC 0.4 mg/mL group showed higher scores than the 0.2 mg/mL group at the 1-month follow-up visit (*p* = 0.042). In the other follow-up evaluations, the bleb height scores showed no significant differences between the two groups. None of the other six parameters showed statistically significant inter-group differences at any of the follow-up visits.

### 3.5. Complications and Additional Procedures

The postoperative complications and additional procedures are shown in Table 5. During the follow-up period, we were unable to detect bleb leak, blebitis, or endophthalmitis. Hypotony was found in two eyes (5.6%), and it was the most frequent of the complications. Hyphema occurred in one patient, and improved without any additional intervention. Choroidal detachment also occurred in one patient. It was observed only at the periphery and was improved after stopping the glaucoma medication that had been used until that time. In all of the cases of complication, there was no significant difference between the two groups. Laser suture lysis and needling were performed in 19 (52.8%) and 9 eyes (25.0%), respectively. 5-fluorouracil (FU) injection and bleb massage were performed in 2 (5.6%) and 15 eyes (41.7%), respectively. No surgical bleb revision or secondary glaucoma surgery was required. In all of the additional procedures, there was no significant difference between the two groups.

## 4. Discussion

We evaluated the changes in IOP and number of glaucoma medications, success rate, and bleb morphology in terms of efficacy. The IOP decreased by an average of 12.50 mg, which corresponded to a decrease of 44.98% compared with the preoperative IOP. The number of glaucoma medications also decreased. This indicates that the trabeculectomy with MMC effectively reduced the IOP in both the 0.2 mg/mL and 0.4 mg/mL MMC groups. We also evaluated the surgical success rate, applying the same three criteria as in the relevant previous studies [6,24,25]. The average success rate was 75.0, 66.7, and 47.2% for criteria A, B, and C, respectively, at the 6-month follow-up visits. As the criteria became more stringent, the success rate decreased. These results are in agreement with previous studies [6,24,25].

Several studies on the success rate of trabeculectomy using MMC have reported various success rates. This might have been because the studies had defined surgical success based on different criteria and the durations of follow-up were also different. In our study, the success rates were 75.0%, 66.7%, and 47.2% at the 6-month visit. Fontana’s study used similar definitions of success to ours [24,25]. They reported that in phakic eyes, the success rates were 62%, 56%, and 46% at 3 years for the criteria ≤18 mmHg and ≥20% IOP reduction, ≤15 mmHg and ≥25% reduction, and ≤12 mmHg and ≥30% reduction, respectively, and that in pseudo-phakic eyes, the rates were 67%, 58%, and 50% at 2 years for the same criteria [24,25]. According to similar criteria, Jampel et al. reported success rates of 72%, 60%, and 44% at 4 years after surgery [6]. Our study showed slightly higher success rates than the relevant previous studies, which might have been because our follow-up period was relatively short.

In the comparison of the two groups with different concentrations of MMC, there was no significant difference in the effect of surgery. However, the concentrations of MMC, although not statistically different in terms of efficacy, showed a tendency for better results with higher MMC concentrations. Previously, Jampel et al. reported that higher concentrations of MMC were associated with surgical success when other factors were adjusted [6]. Mietz and Kriclstein et al. also reported that the use of high concentrations is important [26]. They determined that IOP was lower when 0.5 mg/mL rather than 0.2 mg/mL of MMC was used. Kitazawa et al. reported that trabeculectomy with 0.2 mg/mL of MMC was more successful than with 0.02 mg/mL [27]. In contrast, Sanders et al. reported that surgery with MMC 0.2 mg/mL and 0.4 mg/mL showed the same results [14].

In our study, one of the risk factors for surgical failure was low preoperative IOP. This was a risk factor for all of the criteria (A, B, and C) in the univariate analysis and for criterion C in the multivariate analysis. It is well-known that the absolute value of change in IOP after surgery might not be large when the preoperative IOP is low [28,29]. In general, eyes with a higher preoperative IOP had a higher target IOP than eyes with a lower preoperative IOP. Therefore, we defined success based on not only the absolute value of IOP reduction but also the percentage reduction of IOP. Nevertheless, our study showed that low preoperative IOP was associated with surgical failure. Similarly to our findings, Jampel et al. showed that European-derived race, use of MMC, higher concentrations of MMC, and higher preoperative IOP are associated with success [6].

MMP-9 was also associated with failure when surgical success was defined according to criteria B. To investigate the effect of MMP on surgical outcome, Helin-Toiviainen et al., similarly to our study, analyzed MMP by obtaining conjunctival tissue from 25 patients undergoing deep sclerotomy surgery [30]. In their study, the surgery group showed a higher density of MMP staining than the control group, but there was no statistically significant difference between the success and failure groups. In their study, they also showed that MMP is associated with the length of use of topical pilocarpine treatment. Another report has also shown that MMP increases in POAG patients after using glaucoma medications [31]. Our study found that a higher density of MMP-9 might be associated with surgical failure. However, the results should be interpreted in consideration of the fact that other factors such as tissue inhibitor of metalloproteinase (TIMP) and the duration of glaucoma medication use might affect each other. MMP9 was found to be relevant only for criterion B. When different criteria (A, B, and C) were used to define success, the subjects identified as having success were different. Therefore, in the univariate analysis, the factors showing a *p*-value of 0.2 or less came out differently, and it is estimated that different results were also produced in the multivariate analysis with these factors. It is therefore considered necessary to analyze the effects of MMP9 by targeting a larger number of them.

Laser suture lysis was one of the risk factors in the univariate analysis for criteria B, even though it was not a significant risk factor in the multivariate analysis. Laser suture lysis is commonly performed after trabeculectomy to improve bleb function. Previously, Fontana et al. showed that laser suture lysis was associated with surgical success in pseudo-phakic eyes; however, in another study, they reported that laser suture lysis was associated with surgical failure in phakic eyes [24,25]. They suggested that the cause of this contradiction was that the conjunctival tissue was so thick that finding a suture was difficult, which can make suture lysis unsuccessful. In addition, they noted that even if they had succeeded in the suture lysis, there was no additional IOP reduction, because the trabeculectomy was already scarring [25]. In general, if early postoperative IOP is poorly controlled, suture lysis is performed. Therefore, even if suture lysis is performed, the result of surgery might not be better than for patients showing well-controlled early-postoperative IOP.

Bleb grading is one of the methods we utilized to evaluate the efficacy of surgery. Bleb is a visible part that is closely related to surgical complications. For example, the presence of a thin or leaking bleb is considered to be a risk factor for blebitis or endophthalmitis [32,33]. It is also one of the clinical indicators of the long-term success of trabeculectomy. It is associated with IOP control, and observing it in detail makes it possible to predict functional surgical outcomes [34,35]. In our study, we did not observe any significant differences other than bleb height at the 1-month visit after surgery. At this visit, the blebs of the MMC 0.4 mg/mL group were significantly higher than those of the 0.2 mg/mL group. At the 6-month visit, even though the MMC 0.4 mg/mL group showed a higher bleb than the 0.2 mg/mL group, there was no significant inter-group difference. Whereas it is still unclear whether there is any correlation between bleb height and IOP control, several studies have shown that successful blebs are of low height [36,37,38,39]. Singh et al. reported that failed blebs were mostly low, and Narita et al. showed that the majority of successful blebs were of moderate or high height [38,39].

In the present study, hypotony was the most common complication, occurring in 2 of 36 eyes (5.6%). In our study, none of the eyes with hypotony showed hypotonic retinopathy. Previously, several studies have shown incidence rates of hypotonic retinopathy as low as 3%–13.5% [40,41,42,43,44]. Sunar et al. reported an only 1% incidence rate of hypotonic retinopathy despite administration of 0.5% MMC at high concentrations for 5 min [45]. Singh et al. reported that none of the 54 patients who had received trabeculectomy with 0.4 mg/mL MMC developed hypotonic maculopathy [46]. Blebitis and endophthalmitis have been reported to occur at average rates, respectively, of 6% and 0.8%–1.3% per year after trabeculectomy [33,47,48]. In our study, neither blebits nor endophthalmitis occurred. All of the surgeons in our study were experienced glaucoma specialists, which may have affected the low rate of complications. The relatively short-term follow-up period and small number of subjects might also be reasons for this result. Among the additional procedures, laser suture lysis was most commonly performed in our study. Nineteen of 36 patients (52.8%) received laser suture lysis. None of the additional procedures, including laser suture lysis, needling, bleb massage, and 5-FU injection, were performed more frequently in one group than in the other. This suggests that the postoperative course was similar between the two groups.

The limitations of this study are as follows. First, the follow-up period was relatively short, and thus, we could not consider long-term surgical prognoses. Second, in order to more reliably demonstrate the safety of surgical complications that occur infrequently, it is necessary to analyze a larger number of target patients. Third, the amount of sample tissue was insufficient, so we could not determine the amounts of MMP in tissues. Further study with a larger number of cases and sufficient amounts of sample as well as quantitative evaluation, using, for example, the reverse transcription polymerase chain reaction (RT-PCR) method, is needed. Fourth, it is unclear how factors other than MMC concentration, such as site and duration of MMC application, affect surgical results. In this study, those factors were the same in all patients because we had aimed to compare the two groups only according to different concentrations of MMC.

In conclusion, IOP was well controlled in groups of patients administered MMC at 0.2 mg/mL or 0.4 mg/mL, and the rate of complications was low in both groups. The differences in efficacy and safety between 0.2 mg/mL and 0.4 mg/mL of MMC administration in cases of trabeculectomy were not significant. These results suggest that 0.2 mg/mL and 0.4 mg/mL of MMC are useful for trabeculectomy, which reduces IOP both effectively and safely. Because we cannot say that whether either 0.2 mg/mL or 0.4 mg/mL of MMC is better than the other, it is necessary to determine the appropriate concentration of MMC in consideration of the individual patient′s condition.

## Figures and Tables

**Figure 1 jcm-10-00059-f001:**
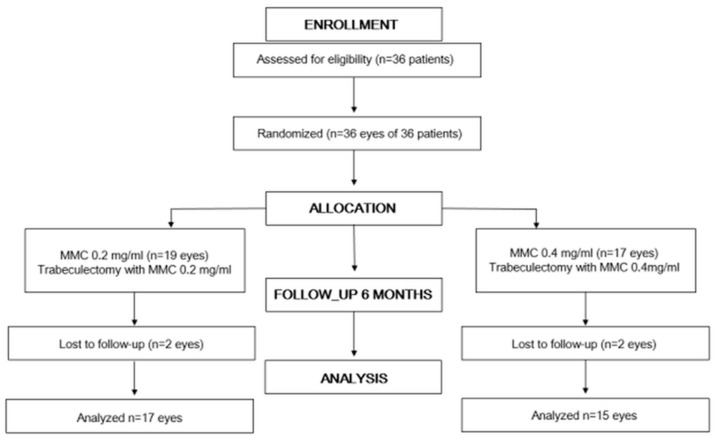
Flow chart according to Consolidated Standards of Reporting Trials (CONSORT).

**Figure 2 jcm-10-00059-f002:**
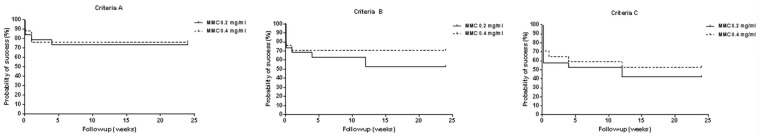
Kaplan–Meier estimates for surgical success according to three criteria for success. Criteria A (intraocular pressure (IOP) ≤ 18 mmHg and IOP reduction ≥ 20%), B (IOP ≤ 15 mmHg and IOP reduction ≥ 25%), C (IOP ≤ 12 mmHg and IOP reduction ≥ 30%).

**Table 1 jcm-10-00059-t001:** Demographics and clinical characteristics.

	Total (*n* = 36)	Group 1 (MMC 0.2 mg/mL) (*n* = 19)	Group 2 (MMC 0.4 mg/mL) (*n* = 17)
Age (years), mean ± SD	58.79 ± 15.86	53.83 ± 15.29	64.33 ± 15.02
Sex, *n* (%)			
Male	23 (63.9)	10 (52.6)	13 (76.5)
Female	13 (36.1)	9 (47.4)	4 (23.5)
Diabetes, *n* (%)	7 (19.4)	5 (26.3)	2 (11.8)
Systemic hypertension, *n* (%)	15 (41.7)	6 (31.6)	9 (52.9)
Rheumatic disease, *n* (%)	2 (5.6)	1 (5.3)	1 (5.9)
Surgeon, *n* (%)			
K.H.P.	20 (55.6)	9 (47.4)	11 (64.7)
J.W.J.	5 (13.9)	3 (15.8)	2 (11.8)
Y.K.K.	11 (30.6)	7 (36.8)	4 (23.5)
Type of glaucoma, *n* (%)			
Primary open-angle glaucoma	19 (52.8)	11 (57.9)	8 (47.1)
Primary angle-closure glaucoma	4 (11.1)	2 (10.5)	2 (11.8)
Pseudo-exfoliation glaucoma	4 (11.1)	2 (10.5)	2 (11.8)
Secondary glaucoma	9 (25.0)	4 (21.1)	5 (29.4)
Previous laser history, *n* (%)	7 (19.4)	4 (21.1)	3 (17.6)
Previous cataract operation history, *n* (%)	7 (19.4)	4 (21.1)	3 (17.6)
Central corneal thickness (um), mean ± SD	532.75 ± 42.34	529.22 ± 33.86	537.29 ± 52.32
Axial length (mm), mean ± SD	24.15 ± 1.77	23.98 ± 1.14	24.33 ± 2.30
Preoperative BCVA, mean ± SD	0.37 ± 0.37	0.38 ± 0.48	0.37 ± 0.21
Preoperative IOP (mmHg), mean ± SD	24.72 ± 8.64	26.34 ± 10.00	22.91 ± 6.63
Preoperative medications, mean ± SD	2.94 ± 0.92	3.11 ± 0.81	2.76 ± 1.03
MD (decibel), mean ± SD	−17.65 ± 10.54	−18.34 ± 10.41	−16.87 ± 10.96
PSD (decibel), mean ± SD	7.56 ± 3.66	7.93 ± 3.34	7.13 ± 4.04
VFI mean ± SD	48.20 ± 34.68	46.58 ± 33.37	50.13 ± 37.19

MMC; mitomycin C, SD; standard deviation, BCVA; best-corrected visual acuity, IOP; intraocular pressure, MD; mean deviation, PSD; pattern standard deviation, VFI; visual field index. ^a^ Mann–Whitney test. ^b^ Fisher’s exact tests.

**Table 2 jcm-10-00059-t002:** Comparison of intraocular pressure and number of glaucoma medications between MMC 0.2 mg/mL group and MMC 0.4 mg/mL group.

	Total (*n* = 36)	Group 1 (MMC 0.2 mg/mL) (*n* = 19)	Group 2 (MMC 0.4 mg/mL) (*n* = 17)	*p-*Value
IOP (mmHg), mean ± SD				
Preoperative	24.72 ± 8.64	26.34 ± 10.00	22.91 ± 6.63	0.230 ^a^
Postoperative 1 day	14.01 ± 8.84	16.24 ± 10.45	11.53 ± 5.99	0.112 ^a^
Postoperative 1 week	12.26 ± 5.55	12.87 ± 6.74	11.59 ± 3.92	0.498 ^a^
Postoperative 1 month	12.47 ± 4.63	11.53 ± 4.50	13.53 ± 4.67	0.199 ^a^
Postoperative 3 months	12.03 ± 3.30	11.94 ± 4.03	12.12 ± 2.50	0.879 ^a^
Postoperative 6 months	12.88 ± 4.63	12.71 ± 5.69	13.07 ± 3.22	0.830 ^a^
IOP reduction (mmHg), mean ± SD				
Postoperative 1 day	10.71 ± 13.17	10.11 ± 15.51	11.38 ± 10.40	0.776 ^a^
Postoperative 1 week	12.46 ± 11.47	13.47 ± 13.40	11.32 ± 10.01	0.582 ^a^
Postoperative 1 month	12.25 ± 9.51	14.82 ± 10.01	9.38 ± 8.27	0.087 ^a^
Postoperative 3 months	12.62 ± 8.76	14.44 ± 9.96	10.79 ± 7.21	0.230 ^a^
Postoperative 6 months	12.50 ± 8.42	14.15 ± 9.51	10.63 ± 6.83	0.245 ^a^
Percentage of IOP reductio*n* (%), mean ± SD				
Postoperative 1 day	34.29 ± 47.18	26.70 ± 53.07	42.77 ± 39.45	0.315 ^a^
Postoperative 1 week	40.05 ± 45.27	37.49 ± 56.10	42.91 ± 30.44	0.725 ^a^
Postoperative 1 month	43.42 ± 26.64	50.11 ± 23.33	35.93 ± 28.76	0.112 ^a^
Postoperative 3 months	45.39 ± 22.89	48.22 ± 25.39	42.57 ± 20.48	0.480 ^a^
Postoperative 6 months	44.98 ± 20.92	48.53 ± 20.80	40.93 ± 21.02	0.311 ^a^
Number of glaucoma medications, mean ± SD				
Preoperative	2.94 ± 0.92	3.11 ± 0.81	2.76 ± 1.03	0.276 ^a^
Postoperative 1 day	0.14 ± 0.42	0.26 ± 0.56	0	0.056 ^a^
Postoperative 1 week	0.19 ± 0.47	0.26 ± 0.56	0.12 ± 0.33	0.358 ^a^
Postoperative 1 month	0.33 ± 0.59	0.32 ± 0.58	0.35 ± 0.61	0.852 ^a^
Postoperative 3 months	0.56 ± 0.82	0.59 ± 0.71	0.53 ± 0.94	0.839 ^a^
Postoperative_6 months	0.53 ± 0.72	0.59 ± 0.71	0.47 ± 0.74	0.640 ^a^

MMC; mitomycin C, IOP; intraocular pressure, SD; standard deviation. ^a^ Mann–Whitney test.

**Table 3 jcm-10-00059-t003:** Univariate and multivariate Cox’s proportional hazard model data for prediction of failure.

	Criterion A (IOP ≤18 mmHg and ≥20%)			Criterion B (IOP ≤15 mmHg and ≥25%)			Criterion C (IOP ≤12 mmHg and ≥30%)		
**Univariate Analysis**									
	HR	95% CI	*p*-value	HR	95% CI	*p*-value	HR	95% CI	*p*-value
Age	0.997	0.958–1.038	0.877 ^a^	1.016	0.980–1.052	0.390 ^a^	1.025	0.992–1.058	0.134 ^a^
Sex	0.937	0.234–3.752	0.927 ^a^	0.992	0.332–2.960	0.988 ^a^	1.340	0.539–3.334	0.529 ^a^
MMC concentration	0.883	0.237–3.290	0.853 ^a^	0.606	0.203–1.810	0.370 ^a^	0.781	0.314–1.942	0.595 ^a^
Surgeon	0.612	0.257–1.459	0.268 ^a^	0.711	0.374–1.354	0.299 ^a^	1.073	0.649–1.774	0.783 ^a^
Diabetes	1.694	0.490–7.866	0.340 ^a^	1.090	0.303–3.920	0.895 ^a^	1.175	0.390–3.544	0.775 ^a^
Systemic hypertension	0.694	0.173–2.774	0.605 ^a^	0.545	0.171–1.740	0.306 ^a^	1.029	0.414–2.558	0.952 ^a^
Rheumatic disease	0.045	0.000–8587.18	0.618 ^a^	1.468	0.192–11.243	0.712 ^a^	0.993	0.133–7.441	0.995 ^a^
Type of glaucoma	0.492	0.216–1.122	0.092 ^a^	0.771	0.485–1.224	0.270 ^a^	0.829	0.572–1.202	0.322 ^a^
Previous laser treatment	0.478	0.060–3.826	0.487 ^a^	0.663	0.148–2.964	0.591 ^a^	0.754	0.229–2.692	0.700 ^a^
Previous cataract op	1.288	0.267–6.208	0.753 ^a^	1.122	0.313–4.024	0.859 ^a^	1.429	0.515–3.969	0.493 ^a^
CCT	1.003	0.985–1.021	0.771 ^a^	0.997	0.984–1.010	0.634 ^a^	0.998	0.987–1.009	0.736 ^a^
AXL	1.088	0.768–1.540	0.636 ^a^	1.028	0.757–1.396	0.860 ^a^	0.890	0.634–1.249	0.500 ^a^
Preoperative BCVA	2.309	0.527–10.126	0.267 ^a^	1.507	0.424–5.358	0.527 ^a^	1.157	0.354–3.786	0.809 ^a^
Preoperative IOP	0.858	0.764–0.964	0.010 ^a^	0.910	0.842–0.984	0.018 ^a^	0.936	0.881–0.995	0.033 ^a^
VF MD	1.005	0.944–1.070	0.866 ^a^	1.005	0.956–1.056	0.852 ^a^	1.006	0.965–1.050	0.765 ^a^
VF PSD	0.979	0.820–1.170	0.818 ^a^	1.028	0.889–1.189	0.709 ^a^	1.033	0.912–1.169	0.611 ^a^
VF VFI	0.999	0.980–1.019	0.948 ^a^	1.000	0.985–1.016	0.971 ^a^	1.001	0.988–1.014	0.854 ^a^
Laser suture lysis	1.923	0.480–7.698	0.355 ^a^	3.895	1.081–14.036	0.038 ^a^	2.276	0.086–6.022	0.098 ^a^
MMP-1 staining	1.526	0.184–12.680	0.696 ^a^	2.922	0.373–22.871	0.307 ^a^	4.017	0.522–30.896	0.182 ^a^
MMP-2 staining	1.250	0.243–6.443	0.790 ^a^	2.570	0.553–11.935	0.228 ^a^	1.975	0.549–7.097	0.297 ^a^
MMP-3 staining	1.123	0.131–9.627	0.916 ^a^	2.473	0.313–19.558	0.391 ^a^	3.190	0.410–24.837	0.268 ^a^
MMP-9 staining	5.464	0.635–47.030	0.122 ^a^	5.556	1.147–26.922	0.033 ^a^	1.835	0.595–5.658	0.290 ^a^
**Multivariate Analysis**									
Age							1.018	0.984–1.053	0.302 ^b^
Type of glaucoma	0.521	0.202–1.347	0.179 ^b^						
Preoperative IOP	0.897	0.776–1.036	0.138 ^b^	0.926	0.837–1.025	0.137 ^b^	0.936	0.881–0.995	0.033 ^b^
Laser suture lysis							2.286	0.844–6.192	0.104 ^b^
MMP-9 stating	5.464	0.635–47.030	0.122 ^b^	5.556	1.147–26.922	0.033 ^b^			

HR; hazard ratio, CI; confidential interval, MMC; mitomycin C, CCT; central corneal thickness, AXL; axial length, BCVA; best-corrected visual acuity, IOP; intraocular pressure, VF; visual field, MD; mean deviation, PSD; pattern standard deviation, VFI; visual field index, MMP; matrix metalloproteinase, CI; confidential interval. ^a^ Univariate logistic regression analysis, ^b^ Multivariate logistic regression analysis.

**Table 4 jcm-10-00059-t004:** Bleb morphology during follow-up period.

	Total (*n* = 36)	Group 1 (MMC 0.2 mg/mL) (*n* = 19)	Group 2 (MMC 0.4 mg/mL) (*n* = 17)	*p*-Value
Bleb area: central				
1 week	2.78 ± 1.22	2.79 ± 1.25	2.77 ± 1.24	0.973 ^a^
1 month	2.73 ± 1.08	2.82 ± 1.25	2.64 ± 0.92	0.703 ^a^
3 months	2.77 ± 1.02	2.92 ± 1.17	2.60 ± 0.84	0.469 ^a^
6 months	2.76 ± 1.00	2.73 ± 1.19	2.80 ± 0.79	0.870 ^a^
Bleb area: maximal				
1 week	3.30 ± 1.07	3.36 ± 1.22	3.23 ± 0.93	0.763 ^a^
1 month	3.27 ± 0.98	3.18 ± 1.08	3.36 ± 0.92	0.676 ^a^
3 months	3.32 ± 0.89	3.42 ± 1.08	3.20 ± 0.63	0.584 ^a^
6 months	3.43 ± 0.81	3.55 ± 1.04	3.30 ± 0.48	0.502 ^a^
Bleb height				
1 week	1.33 ± 0.48	1.43 ± 0.51	1.23 ± 0.44	0.291 ^a^
1 month	1.59 ± 0.73	1.27 ± 0.47	1.91 ± 0.83	0.042 ^a^
3 months	1.91 ± 1.02	1.67 ± 0.49	2.20 ± 1.40	0.230 ^a^
6 months	2.05 ± 1.02	1.82 ± 0.75	2.30 ± 1.25	0.293 ^a^
Bleb vascularity: central				
1 week	2.15 ± 0.86	2.36 ± 0.63	1.92 ± 1.04	0.198 ^a^
1 month	1.86 ± 0.83	2.18 ± 0.87	1.55 ± 0.69	0.073 ^a^
3 months	1.50 ± 0.60	1.58 ± 0.67	1.40 ± 0.52	0.477 ^a^
6 months	1.52 ± 0.51	1.55 ± 0.52	1.50 ± 0.53	0.845 ^a^
Bleb vascularity: peripheral				
1 week	2.89 ± 0.64	2.86 ± 0.66	2.92 ± 0.64	0.795 ^a^
1 month	2.55 ± 0.80	2.64 ± 0.81	2.45 ± 0.82	0.606 ^a^
3 months	2.18 ± 0.50	2.17 ± 0.58	2.20 ± 0.42	0.877 ^a^
6 months	2.10 ± 0.30	2.09 ± 0.30	2.10 ± 0.32	0.947 ^a^
Bleb vascularity: non-bleb				
1 week	2.30 ± 0.54	2.14 ± 0.54	2.46 ± 0.52	0.129 ^a^
1 month	2.09 ± 0.68	2.18 ± 0.75	2.00 ± 0.63	0.546 ^a^
3 months	1.86 ± 0.47	1.75 ± 0.45	2.00 ± 0.47	0.222 ^a^
6 months	2.05 ± 0.38	2.00 ± 0.45	2.10 ± 0.32	0.559 ^a^
Subconjunctival hemorrhage				
1 week	0.48 ± 0.51	0.36 ± 0.50	0.62 ± 0.51	0.193 ^a^
1 month	0.05 ± 0.21	0.00	0.09 ± 0.30	0.329 ^a^
3 months	0.05 ± 0.21	0.08 ± 0.30	0	0.339 ^a^
6 months	0	0	0	N/A

MMC; mitomycin C, N/A; not applicable. ^a^ Mann–Whitney test.

**Table 5 jcm-10-00059-t005:** Postoperative complications and additional procedures.

	Total (*n* = 36)	Group 1 (MMC 0.2 mg/mL) (*n* = 19)	Group 2 (MMC 0.4 mg/mL) (*n* = 17)	*p*-Value
**Complications**				
Hyphema, *n* (%)	1 (2.8)	0	1 (5.9)	0.472 ^a^
Hypotony, *n* (%)	2 (5.6)	2 (10.5)	0	0.487
Bleb leak, *n* (%)	0	0	0	N/A
Blebitis, *n* (%)	0	0	0	N/A
Endophthalmitis, *n* (%)	0	0	0	N/A
Choroidal detachment, *n* (%)	1 (2.8)	1 (5.3)	0	1.000 ^a^
**Additional procedures**				
Laser suture lysis, *n* (%)	19 (52.8)	12 (63.2)	7 (41.2)	0.316 ^a^
Bleb needling, *n* (%)	9 (25.0)	5 (26.3)	4 (23.5)	1.000 ^a^
Bleb massage, *n* (%)	15 (78.9)	10 (52.6)	5 (29.4)	0.192 ^a^
5 FU injection, *n* (%)	2 (10.5)	0	2 (11.8)	0.216 ^a^
Additional surgery, *n* (%)(bleb revision or secondary glaucoma surgery)	0	0	0	N/A

MMC; mitomycin C, 5-FU; 5-fluorouracil, N/A; not applicable. ^a^ Fisher’s exact tests.

## Data Availability

The data presented in this study are available in this article.

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
