# Peer review of "Comparison of the Efficacy and Safety of Trabeculectomy with Mitomycin C According to Concentration: A Prospective Randomized Clinical Trial"

_jcm, 2020, doi:10.3390/jcm10010059_

Round 1

Reviewer 1 Report

The current study is of interest for the field of Glaucoma treatment. The methodology, especially the statistics and design are sound. My only concern it that the plagiarism check showed 35% similarity with previously published data.. thus, there's a need for a rephrasing and check by an English native Speaker. 

Lines 27-29: Many key sentences lack references, the authors are encouraged to add the adequate reference(s).

Lines 32-33 and lines 36, 37: the authors have separated the objectives since they added every objective in a different paragraph. The authors are encouraged to re-organize these objectives into one single paragraph preferably at the end of the introduction..

Line 77: ‘will be’ I believe it is a type and can be replaced with was.  

Lines 151-154: he authors cite the exact values in the text (24.72 (8.64)) despite being displayed in Table 2. The authors are encouraged to decrease this overlap with the Table and instead summarize these findings. Readers can retrieve this information from the Table.

Table 4: MMP9 staining was significantly associated with failure in criteria B but not A and C.. this is interesting, the authors could discuss thoroughly this idea as it is of interest for the readers.

Table 5. I believe that a chi-square test was used to assess the significance of the frequencies between group 1 vs.2. Unfortunately, this information is missing and need to be added in the statistical analysis section.

Author Response

Reviewer 1

The current study is of interest for the field of Glaucoma treatment. The methodology, especially the statistics and design are sound. My only concern it that the plagiarism check showed 35% similarity with previously published data.. thus, there's a need for a rephrasing and check by an English native Speaker.

  • This has been revised in English by experts according to your advice.

Lines 27-29: Many key sentences lack references, the authors are encouraged to add the adequate reference(s).

  • We added the appropriate reference(s) (Lines 27-29).

Lines 32-33 and lines 36, 37: the authors have separated the objectives since they added every objective in a different paragraph. The authors are encouraged to re-organize these objectives into one single paragraph preferably at the end of the introduction..

  • As you recommended, we have rewritten it. (Lines 27-38).

Line 77: ‘will be’ I believe it is a type and can be replaced with was.

  • We modified “will be” to “was”. (Lines 76).

Lines 151-154: he authors cite the exact values in the text (24.72 (8.64)) despite being displayed in Table 2. The authors are encouraged to decrease this overlap with the Table and instead summarize these findings. Readers can retrieve this information from the Table.

  • As you recommended, we deleted the exact values in the text (Lines 145-150).

Table 4: MMP9 staining was significantly associated with failure in criteria B but not A and C.. this is interesting, the authors could discuss thoroughly this idea as it is of interest for the readers.

  • MMP9 was found to be relevant only for criterion B. When different criteria (A, B, and C) were used to define success, the number and composition of subjects identified as having success would have been different. Therefore, in the univariate analysis, the factors showing a p-value of 0.2 or less came out differently, and it is estimated that different results were also produced in the multivariate analysis with these factors. For this reason, it is considered necessary to analyze the effects of MMP9 by targeting a larger number of them. We described this in the discussion section (Lines 218-227).

Table 5. I believe that a chi-square test was used to assess the significance of the frequencies between group 1 vs.2. Unfortunately, this information is missing and need to be added in the statistical analysis section.

  • We used “Fisher’s exact tests” instead of “chi-square test” because we had a small number of patients. We added this in the statistical analysis section (Lines 118-125).

Reviewer 2 Report

In this manuscript the authors present results of a prospective, randomized, interventional study on the use of different concentrations of mitomycin c in trabeculectomy procedures. It is reported that this study was registered in a national registry after finishing the study. This is rather surprising and unusual. For many years the usual standard for such a type of interventional study is to have it registered before initiation and to use one of the internationally recognized resources for registration. This has been for many years an essential prerequisite for publication in international journals.

It is striking that the authors were very optimistic when calculating the number of necessary individual necessary for the designed study approach. This clearly contributes to the calculation of rather low numbers of necessary individuals. I cannot share the set preconditions. According to my understanding the calculation would have to be adjusted in a way that is leading to a much higher number of individuals. The calculation was made for treatment efficacy only and not for safety. It is clear that for any conclusion on safety would need much higher numbers of individuals. In addition to the shortcomings in study size, the short follow-up of only 6 months is to short to make any resilient conclusion on safety and efficacy in a procedure like this. The available literature clearly demonstrates that most authors share this opinion.

It is striking that that there is a more than 10-years difference in the mean age of patients in the two compared groups. The difference was statistically significant. This is very surprising. With regard to these figures doubt about the randomization process is lurking in my mind.

Author Response

Reviewer 2

In this manuscript the authors present results of a prospective, randomized, interventional study on the use of different concentrations of mitomycin c in trabeculectomy procedures. It is reported that this study was registered in a national registry after finishing the study. This is rather surprising and unusual. For many years the usual standard for such a type of interventional study is to have it registered before initiation and to use one of the internationally recognized resources for registration. This has been for many years an essential prerequisite for publication in international journals.

  • As mentioned in the method section, our research was conducted with IRB approval prior to commencement. However, we belatedly recognized that there are some international journals that only accept studies registered in a specific national registry. Therefore, it was registered in the national registry after the study was completed so that it would not be a problem for future submissions to international journals. Thus, we don't think that this means the research process was unethical.

It is striking that the authors were very optimistic when calculating the number of necessary individual necessary for the designed study approach. This clearly contributes to the calculation of rather low numbers of necessary individuals. I cannot share the set preconditions. According to my understanding the calculation would have to be adjusted in a way that is leading to a much higher number of individuals. The calculation was made for treatment efficacy only and not for safety. It is clear that for any conclusion on safety would need much higher numbers of individuals. In addition to the shortcomings in study size, the short follow-up of only 6 months is to short to make any resilient conclusion on safety and efficacy in a procedure like this. The available literature clearly demonstrates that most authors share this opinion.

  • Of course, it is ideal to enroll as many patients as possible, but since we had to prospectively recruit patients and collect specimens during surgery, we aimed at the minimum required number. In this study, we compared both the efficacy and safety aspects of the postoperative results according to the MMC concentration. Therefore, the minimum number of patients required may vary depending on which factor is used. Above all, we thought that MMC 0.2 mg/ml and 0.4 mg/ml showed a difference in the inhibition of wound healing, and in this regard, the minimum number that would show a significant difference in the score for the degree of vascularization in the center of the filtering bleb was calculated. We have described in detail the method for calculating this minimum number of subjects in the Methods section (Lines 127-130).

As you mentioned, among the existing studies that analyzed the course after trabeculectomy, many studies were analyzed with a follow-up period of at least 1 year. Even though the follow up period of our study was shorter than those of previous studies, we used more stringent criteria. This may be why our study showed lower success rates than previous studies. We described this in the Discussion section (Lines 197, 199-205). Since this study’s purpose was to evaluate the difference in the success rates between the 2 groups, it is more important to focus on the difference between the two groups rather than comparison of the success rates with the previous studies.

In addition, in order to more reliably demonstrate the safety of surgical complications that occur infrequently, it is necessary to analyze a larger number of target patients.

We agree that, as you pointed out, the results of long-term follow-up with a larger number of subjects will be needed in the future. We described this in the discussion section (Lines 253-257).

It is striking that that there is a more than 10-years difference in the mean age of patients in the two compared groups. The difference was statistically significant. This is very surprising. With regard to these figures doubt about the randomization process is lurking in my mind.

  • We conducted this study as a research project supported by the Ministry of Food and Drug Safety in Korea, and for the randomization process, we applied for randomization at the Medical Research Cooperation Center of Seoul National University Hospital, which specializes in this method. The detailed method is described in the method section. (Lines 70-76)

As you pointed out, it is surprising that the difference in mean age in the two compared groups was more than 10 years. This is probably due to the small number of patients in the two groups. In addition, reviewer 3 commented that it is not necessary to statistically compare baselines values of two groups composed through randomization. Thus, we removed the p-values in Table 1 in the revised manuscript.

Reviewer 3 Report

In this study, Seol et al. invesigate efficacy and safety of trabeculectomy with mitomycin C with different concentrations. I congratulate the authors for conducting this study. The authors acknowledge the limitations of the study, which is important as to how this study should be interpret.

I just have one comment. Why do the authors deem it necessary to statistically compare baseline values of two groups composed through randomization? This is controversial (doi: 10.1177/1741826711421688 ) and the rationale should be explained. Otherwise, consider omitting such p-values.

Author Response

Reviewer 3

In this study, Seol et al. invesigate efficacy and safety of trabeculectomy with mitomycin C with different concentrations. I congratulate the authors for conducting this study. The authors acknowledge the limitations of the study, which is important as to how this study should be interpret.

I just have one comment. Why do the authors deem it necessary to statistically compare baseline values of two groups composed through randomization? This is controversial (doi: 10.1177/1741826711421688 ) and the rationale should be explained. Otherwise, consider omitting such p-values.

  • We removed the p-values in the Table 1.

Round 2

Reviewer 2 Report

Concerning ethics: It is against international standards to have an interventional human study registered after finishing the study. If this would be acceptable the door would be open for any hidden studies that would then be selected after finishing for registration if appropriate. I recognize that the national registry in the country of the authors accepted this but it is clearly against good laboratory practice.

Concerning calculation of the number of necessary individuals: The authors state, that “Of course, it is ideal to enrol as many patients as possible…”. This is definitely not the point: To include as many patients as possible would ethically not correct! What has to be done is to calculate the necessary numbers on realistic assumptions. I do not see, that this was done.

The arguments of the authors concerning the critics on the short follow-up is not convincing. Comparison of two groups cannot substitute the need for longer observation.

Concerning baseline age difference in the compared groups: The scepticism about the randomization process remains. It is improbable that such a difference occurs. In addition it is inappropriate to delete the message of a statistical difference of baseline age in the final version of the manuscript, as reported by the authors. And the given rationale that, “it is not necessary to statistically compare baselines (sic) values of two groups composed through randomization”, cannot be shared and will certainly not be shared by scientific communities. My impression is that the authors, by deleting this message, would like to hide a fundamental lack of the randomization process.

Over all, the responses of the authors demonstrate a fundamental unreasonableness against the risen ethical and study design concerns.

Author Response

Response to reviewer’s comments

Concerning ethics: It is against international standards to have an interventional human study registered after finishing the study. If this would be acceptable the door would be open for any hidden studies that would then be selected after finishing for registration if appropriate. I recognize that the national registry in the country of the authors accepted this but it is clearly against good laboratory practice.

-> We think that the explanation that we gave in response to your question is quite misleading; therefore, we will explain it further. This does not mean that our country accepted the study after registration. The interventional human study was registered in the domestic IRB before the start of the study. However, because it is required that the study be registered on a specific international site (cris.nih.go.kr) in order to be able to submit it to international journals for publication, we therefore registered the study on this site after finishing it. As written in method, after explaining the method, including the benefits and risks of the procedure, informed consent was obtained from all of the patients. The study was conducted in accordance with the Declaration of Helsinki and CONSORT guidelines. And this study was approved by the Institutional Review Board (IRB) of Seoul National University Hospital. Therefore, to avoid confusion, we modified the sentence as follows: “It was approved by the Institutional Review Board (IRB) of Seoul National University Hospital (1410-081-618) on 09/01/2014. This trial was registered at cris.nih.go.kr on 01/07/2019 (KCT0004108)” (Line 44-45). Or, if the international site registration is not required in this journal, we can remove the phrase as follows: “It was approved by the Institutional Review Board (IRB) of Seoul National University Hospital (1410-081-618) on 09/01/2014.”

Concerning calculation of the number of necessary individuals: The authors state, that “Of course, it is ideal to enroll as many patients as possible…”’. This is definitely not the point: To include as many patients as possible would ethically not correct! What has to be done is to calculate the necessary numbers on realistic assumptions. I do not see that this was done.

-> We agree with your comment that it is necessary to calculate the necessary numbers for a realistic assumption. We assumed that MMC 0.2 mg/mL and 0.4 mg/mL would differ in wound healing. In this regard, the minimum number that would show a significant difference in the score for the degree of vascularization in the filtering bleb was calculated. We have described in detail the method for calculating this minimum number of subjects in the Methods section (Line 125-129).

The arguments of the authors concerning the critics of the short follow-up are not convincing. Comparison of the two groups cannot substitute for the need for longer observation.

-> We totally agree with your comment that the comparison of the two groups cannot substitute the need for a longer observation. As you mentioned, it will take a long observation period to confirm the long-term follow-up results, and the results of the 6-month follow-up cannot be replaced. However, not all studies have reported the long-term follow-up results. There are studies that have reported the results of relatively shorter follow-up periods (within 6 months), such as ours (1-5) In addition, Okimoto et al. reported that the early postoperative IOP value is related to the outcome of long-term surgery (6). Although we did not report any long-term results, we believe that our study results are meaningful in that the trabeculectomy outcomes with two different concentrations of MMC were systematically and prospectively analyzed.

1) H Mietz , G K Krieglstein. Short-term clinical results and complications of trabeculectomies performed with mitomycin C using different concentrations. Int Ophthalmol. 1995;19(1):51-6.

2) Michael S Quist , Ninita Brown, Amanda K Bicket, Leon W Herndon. The Short-term Effect of Subtenon Sponge Application Versus Subtenon Irrigation of Mitomycin-C on the Outcomes of Trabeculectomy With Ex-PRESS Glaucoma Filtration Device: A Randomized Trial. J Glaucoma. 2018 Feb;27(2):148-156.

3) R Ramakrishnan , J Michon, A L Robin, R Krishnadas. Safety and efficacy of mitomycin C trabeculectomy in southern India. A short-term pilot study. Ophthalmology. 1993 Nov;100(11):1619-23.

4) Thidarat Leeungurasatien , Preeyanuch Khunsongkiet , Kassara Pathanapitoon , Damrong Wiwatwongwana. Incidence of short-term complications and associated factors after primary trabeculectomy in Chiang Mai University Hospital. Indian J Ophthalmol. 2016 Oct;64(10):737-742.

5) G Picht , Y Mutsch, F Grehn. Follow-up of trabeculectomy. Complications and therapeutic consequence. Ophthalmologe. 2001 Jul;98(7):629-34. 

6) Okimoto S, Kiuchi Y, Akita T, Tanaka J. Using the early postoperative intraocular pressure to predict pressure control after a trabeculectomy. J Glaucoma. 2014 Aug;23(6):410-4. 

Concerning baseline age difference in the compared groups: scepticism about the randomization process remains. It is improbable that such a difference occurs. In addition, it is inappropriate to delete the message of a statistical difference of baseline age in the final version of the manuscript, as reported by the authors. And the given rationale that, “it is not necessary to statistically compare baselines (sic) values of two groups composed through randomization” cannot be shared and will certainly not be shared by scientific communities. My impression is that the authors, by deleting this message, would like to hide a fundamental lack of the randomization process.

-> We did not remove the P value when we first wrote the manuscript. However, we excluded the P value by referring to the suggestions of reviewer 3 and previous studies. Several studies and the CONSORT statement showed that testing for baseline differences between the treatment and control groups in randomized controlled trials (RCTs) is not appropriate (1–4). In addition, Grobbee et al. reported that any baseline differences between the groups under study are by definition due to chance (as long as the randomization was performed correctly) (5). For this reason, we cannot give a definite explanation. However, we do not think that there was a problem in our randomization process because of the age difference between the two groups. We conducted the randomization process following the correct procedure. The study was conducted using electronic case report forms (eCRF), and it is possible to assign randomized patients at http://www.phactax.org. Despite our explanation, we can still add again the P value, if you deem  it necessary.

1) MJ Knol, RHH Groenwold, DE. Grobbee P-values in baseline tables of randomized controlled trials are inappropriate but still common in high impact journals. Eur J Prev Cardiol. 2012 Apr;19(2):231-2.

2) Senn S. Testing for baseline balance in clinical trials. Stat Med 1994; 13: 1715–1726.

3) Roberts C and Torgerson DJ. Understanding controlled trials: baseline imbalance in randomized controlled trials. BMJ 1999; 319: 185.

4) http://www.consort-statement.org/consort-statement/ 13-19—results/item15_baseline-data/. (accessed August 2011)

5) Grobbee DE and Hoes AW. Randomized trials. Clinical epidemiology. Sudbury, MA: Jones and Bartlett Publishers, 2009, pp.270–287.

Overall, the responses of the authors demonstrate a fundamental unreasonableness against the risen ethical and study design concerns.

-> Thank you for your kind concerns. We hope that we have appropriately responded to the raised issues. Regarding ethical issues, as mentioned above, the registration on an international site (cris.nih.go.kr) was done after completing the study. However, please take into account that the study was conducted after obtaining approval from the IRB without any ethical problems. In addition, regarding the study design issues, please consider that the selection of the minimum number of patients is based on the bleb vascularity for the reason mentioned above, and this was determined after a formal consultation with the statistical expert of the institute. As for the age difference between the two groups, there is no way to explain the reason for the difference, but it is clear that randomization was carried out through a fair procedure. Randomized patients can be assigned at http://www.phactax.org.